# Structural basis for the inhibition of HTLV-1 integration inferred from cryo-EM deltaretroviral intasome structures

Michal S. Barski [1,6,8], Teresa Vanzo [1,7,8], Xue Zhi Zhao [2], Steven J. Smith [3], Allison Ballandras-Colas [4], Nora B. Cronin[5], Valerie E. Pye [4], Stephen H. Hughes [3], Terrence R. Burke Jr. [2], Peter Cherepanov [1,4] & Goedele N. Maertens [1✉]

Between 10 and 20 million people worldwide are infected with the human T-cell lymphotropic virus type 1 (HTLV-1). Despite causing life-threatening pathologies there is no therapeutic regimen for this deltaretrovirus. Here, we screened a library of integrase strand transfer inhibitor (INSTI) candidates built around several chemical scaffolds to determine their effectiveness in limiting HTLV-1 infection. Naphthyridines with substituents in position 6 emerged as the most potent compounds against HTLV-1, with XZ450 having highest efficacy in vitro. Using single-particle cryo-electron microscopy we visualised XZ450 as well as the clinical HIV-1 INSTIs raltegravir and bictegravir bound to the active site of the deltaretroviral intasome. The structures reveal subtle differences in the coordination environment of the $Mg^{2+}$ ion pair involved in the interaction with the INSTIs. Our results elucidate the binding of INSTIs to the HTLV-1 intasome and support their use for pre-exposure prophylaxis and possibly future treatment of HTLV-1 infection.

[1] Imperial College London, St. Mary's Hospital, Department of Infectious Disease, Section of Virology, Norfolk Place, London, UK. [2] Chemical Biology Laboratory, Centre for Cancer Research, National Cancer Institute, Frederick, MD, USA. [3] Retroviral Replication Laboratory, Centre for Cancer Research, National Cancer Institute, Frederick, MD, USA. [4] Chromatin Structure & Mobile DNA Laboratory, The Francis Crick Institute, London, UK. [5] LonCEM Facility, The Francis Crick Institute, London, UK. [6] Present address: International Institute of Molecular Mechanisms and Machines, Polish Academy of Sciences, Warsaw, Poland. [7] Present address: Department CIBIO, University of Trento, Povo-Trento, Italy. [8] These authors contributed equally: Michal S. Barski, Teresa Vanzo. ✉email: g.maertens@imperial.ac.uk

Human T-cell lymphotropic virus type 1 (HTLV-1) is a deltaretrovirus and one of the most oncogenic human viruses[1]. It is the causative agent of a severe form of blood cancer called adult T-cell leukaemia/lymphoma (ATLL)[2] and a range of debilitating neuromuscular disorders—HTLV-associated myelopathy/tropical spastic paraparesis (HAM/TSP)[3], which often lead to paralysis. Although only 5–10% of the estimated 10–20 million people[4] infected with HTLV-1 worldwide will develop ATLL[2], a recent report showed that the vast majority of otherwise asymptomatic patients suffer from issues that severely impact their quality of life, including pain, discomfort and depression[5]. 12.7% of HAM/TSP patients reported their quality of life was worse than death[5], based on the Euroqol five dimension questionnaire (EQ-5D).

HTLV research has largely remained in the shadow of HIV research, which has moved at a much faster pace. Like HIV-1, HTLV-1 is a blood-borne pathogen that is transmitted horizontally by contact with infected blood, sexual intercourse and vertically from mother to child during breastfeeding and labour. The likelihood of an HTLV-1 carrier developing ATLL is correlated with exposure early in life—most commonly during breastfeeding or child-birth[6]. The prevailing dogma is that, following the initial infection, HTLV replication is primarily the result of the clonal proliferation of infected cells. However, recent data indicate there is ongoing viral replication, which leads to the generation of additional clones of HTLV-1 infected cells, and increases the proviral load. An increased viral load is correlated with an increased risk of developing HTLV-1-associated malignancy and inflammatory disease[7]. Hence, suppression of additional rounds of viral infection may reduce the risk of malignant transformation. Administering anti-retroviral agents may help prevent HTLV-1 transmission among drug users, serodiscordant couples, organ recipients and pregnant/breastfeeding mothers. Thus, strong arguments can be made for developing HTLV-targeted therapeutics[8]. Integration of a DNA copy of the viral genome is an essential step in the replication of all retroviruses. Integrase strand transfer inhibitors (INSTIs) are among the most effective anti-HIV drugs. Similarly, understanding the integrase (IN) enzyme of HTLV, and how it can be inhibited, will be a critical part of the effort to control HTLV infections. We recently reported that the first- and second-generation HIV INSTIs are effective at inhibiting HTLV-1 strand transfer in vitro and integration in tissue culture[9]. In this report, we have characterised several groups of INSTIs that have been reported to inhibit the replication of HIV[10,11].

The target of the INSTIs is the IN active site engaged with viral DNA (vDNA) ends in the context of the nucleoprotein complex termed the intasome[12,13]. Technical advances in cryogenic electron microscopy (cryo-EM), recently allowed the structural characterisation of HIV-1[14] and a closely related red-capped mangabey simian immunodeficiency virus (SIVrcm)[15] intasomes in complexes with clinical and experimental INSTIs at near-atomic resolution. Currently, no structural information is available on the mode of INSTI binding to the deltaretroviral intasome. We recently determined the cryo-EM structure of the deltaretroviral simian T-cell lymphotropic virus type 1 (STLV-1) intasome bound to its host factor B56γ[16]. Here, in addition to characterising three compounds that block HTLV-1 transmission two- to fivefold more potently than raltegravir (RAL), we report three high-resolution cryo-EM structures of a deltaretroviral intasome bound to three different INSTIs. The 3D reconstructions allow for direct comparison between the binding modes of these inhibitors in lentiviral and deltaretroviral intasomes. Our findings have important implications for establishing therapies targeted at HTLV-1 and provides a framework for understanding the mechanism of INSTI action on the HTLV-1 intasome at the atomic scale.

## Results

**XZ450 is a potent HTLV-1 inhibitor.** We evaluated a library of 32 small molecules collectively comprising five classes of known and candidate INSTIs, including naphthyridines, pyridinones, diketo acids, oxoisoindoles and hydrazines (Supplementary Table 1). Most of the compounds were 4-, 6- and 7-substituted naphthyridines, some of which had previously shown remarkable potency against wild-type and mutant forms of HIV-1 IN, including the Q148H/G140S RAL-resistant and dolutegravir-resistant (G118R, T66I, E92Q, R263K, H51Y or H51/R263K) mutants[10,11]. Long aliphatic or aromatic substituents at the 6-position are able to reach across the active site pocket, and associate with the protein backbone as shown in the crystal structures of PFV[10,11], and cryo-EM reconstructions of HIV-1[14] intasomes. We also included naphthyridines designed to inhibit HIV RNase H that were not previously tested against IN, because the compounds also target a pair of magnesium ions in the active site, which is similar to INSTI binding[17]. Cross inhibition has previously been found with hydrazines and other INSTIs, which effectively blocked herpes simplex virus replication by inhibiting RNase H-like fold containing proteins[18].

We used in vitro strand transfer assays to screen this small molecule library for its ability to inhibit recombinant HTLV-1 IN. The enzyme was incubated with an oligonucleotide vDNA donor mimic in the presence or absence of INSTIs and was allowed to react with a supercoiled plasmid mimicking target DNA. During the screening, the concentration of each compound was adjusted to 634 nM to provide optimal resolution of efficacy between well- and under-performing compounds. Summarised in Fig. 1a, the results show that all of the 6-substituted naphthyridines we tested are more potent inhibitors of HTLV-1 IN compared with RAL. In addition, one 4-substituted naphthyridine (XZ378) and one pyridinone (XZ236) also outperformed RAL.

These observations were corroborated using strand transfer assays to obtain dose–response relationships and $IC_{50}$ values (Fig. 1b and Supplementary Fig. 4). Within this panel of selected compounds, the $IC_{50}$ values were two- to fourfold lower than that of RAL, with the exception of XZ420 and XZ378, which showed no significant improvement over RAL. Overall, compounds XZ450, XZ446 and XZ448 exhibited the best characteristics, displaying $IC_{50}$ values of $120.2 \pm 36.1$ nM, $115.4 \pm 9.5$ nM and $131.9 \pm 29.7$ nM, respectively, compared with $410 \pm 78.5$ nM of RAL (Supplementary Table 2). The three compounds are 4-amino-substituted naphthyridines featuring all of which have an additional extended substitution at 6-position with a 3-(dimethylamino)-3-oxopropyl, phenylsulfonylethyl or hydroxyethylaminooxopropyl group, respectively (Supplementary Table 1).

Six of the most promising compounds were selected for assessment of their efficacy in inhibiting HTLV-1 viral infection. To this end, we employed an MT-2-Jurkat co-culture relying on cell-to-cell-based HTLV-1 transmission, which we successfully used in a previous study[9]. In this assay format, the infection of Jurkat cells is measured via real-time quantitative PCR (qPCR) determination of the proviral load (PVL). In line with our in vitro data, XZ450 and XZ446 were most potent in inhibiting HTLV-1 infection of Jurkat T cells (Fig. 1c, and Supplementary Fig. 5), exhibiting ~2.5- to fivefold improvements in $EC_{50}$ ($1.67 \pm 1.52$ nM and $2.57 \pm 0.81$ nM, respectively) compared with RAL ($6.42 \pm 4.24$ nM)[9]. A block in the integration reaction was confirmed by Alu-qPCR experiments (Supplementary Fig. 6). As shown previously[9], although bictegravir (BIC) is only about twofold more efficient in blocking HTLV-1 integrase strand transfer in vitro, BIC potently blocks HTLV-1 transmission in tissue culture, exhibiting a 20-fold improvement in $EC_{50}$ ($0.3 \pm 0.173$ nM)[9] compared with RAL. We suspect this is owing to a more efficient uptake of BIC than RAL by Jurkat T cells[9].

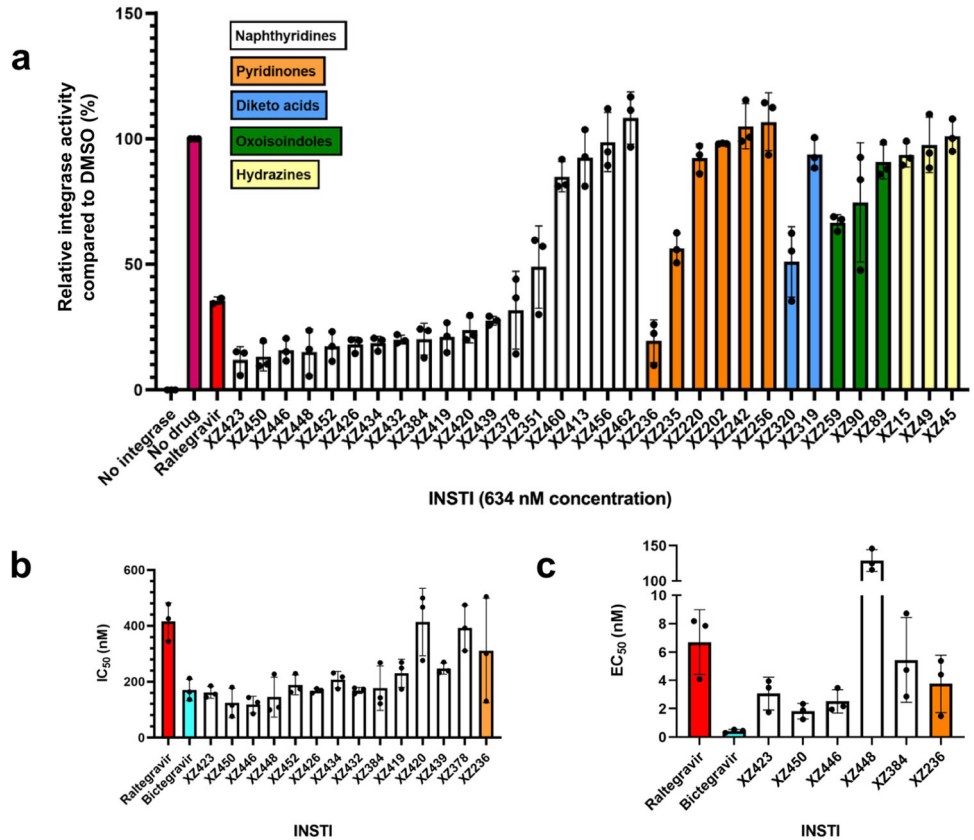

**Fig. 1 Sensitivity of HTLV-1 integrase to different classes of integrase strand transfer inhibitors (INSTIs) in vitro. a** Screening of HTLV-1 IN strand transfer activity in presence of a panel of INSTIs at 634 nM. Results are shown from high to low inhibition activity within each colour-coded INSTI class. Percent inhibition compared with WT HTLV-1 IN in the presence of a vehicle is shown. **b** $IC_{50}$ values were determined for the most promising INSTIs from the naphthyridine and pyridinone classes and compared with previously obtained raltegravir (RAL) and bictegravir (BIC) values[9]. **c** $EC_{50}$ values for INSTIs tested in the MT-2–Jurkat cell-to-cell infection model and compared with previously reported RAL and BIC values[9]. Individual data points, mean values and standard deviations of $n$ = three biologically independent experiments are shown. Exact $IC_{50}$ and $EC_{50}$ values are contained in Supplementary Figs. 4 and 5 and Supplementary Table 2. Source data are provided as a Source Data File.

Interestingly, XZ448, which was more potent than RAL in our in vitro strand transfer reactions exhibited ~20-fold lower activity in blocking HTLV-1 infection compared with RAL (see Discussion).

**Structural studies of STLV-1 intasome:INSTI complexes by cryo-EM.** During our previous work on the apo structure of the deltaretroviral intasome, we identified that the simian variant STLV-1 IN assembled into stable intasomes with greater efficiency than the HTLV-1 counterpart[16]. This is potentially explained by the stronger binding observed between STLV-1 IN and B56γ (Supplementary Fig. 7), the latter being essential for the formation of stable intasomes in vitro[16]. Crucially, HTLV-1 and STLV-1 INs are 83% identical by amino-acid sequence, sharing 100% identity within their active sites. Comparison of the two deltaretroviral intasome cryo-EM structures[16,19] shows a remarkable overlap (Supplementary Fig. 8). Concordantly, similar to HTLV-1 IN, STLV-1 IN is highly sensitive to RAL (Supplementary Fig. 9). STLV-1 has also been reported to cause clinical manifestations in macaques that are similar to those seen in HTLV-1-infected patients. STLV-1-infected macaques are therefore a useful animal model for HTLV-1 infected humans[20]. Collectively, these observations validate STLV-1 IN as an accurate and robust model for HTLV-1 IN structural drug studies.

Our previous work showed that there was a time-dependent disassociation of the STLV-1 intasome, which necessitated quick grid freezing. Low numbers of class-forming particles per

micrograph led to long collection times. This was not conducive to producing multiple intasome:drug complexes at near-atomic resolution. The affinity of the endogenous binding partners for PP2A-B56 is regulated by phosphorylation at the Ser residue following the hydrophobic Leu in the LxxIxE Short Linear Motif (SLiM)[21]. Deltaretroviral INs (SLiM residues [213]LxxIxE[218]) harbour a Pro at this position, we thus investigated whether mutating Pro214 to a phosphomimetic Asp could increase the affinity between IN and B56γ. Moreover, a previous report showed that peptides in which additional acidic residues (Glu) were added immediately following the SLiM sequence, had significantly higher affinity for B56[22]. We thus mutated IN Ala219 into Glu. Approximately twofold more B56γ was recovered in pull-down assays when IN(A219E) was used as a bait compared with WT IN (Supplementary Fig. 10a). In contrast, there was only a minor increase in the amount of B56γ recovered in the pull-down assay with IN(P214D). The IN(A219E) mutant also formed approximately twice as much stable intasome complexes in an electrophoretic mobility shift assay (Supplementary Fig. 10b). This, together with the addition of the INSTI to freshly assembled intasomes considerably improved sample stability (Supplementary Fig. 10c). As expected, the introduction of the A219E mutation in STLV-1 IN did not change its INSTI-binding properties and produced a nearly identical $IC_{50}$ for RAL compared with wild-type STLV-1 IN (Supplementary Fig. 9).

Overall, the improvements in sample stability and optimisation of the graphene oxide (GO) grid support (the latter increasing

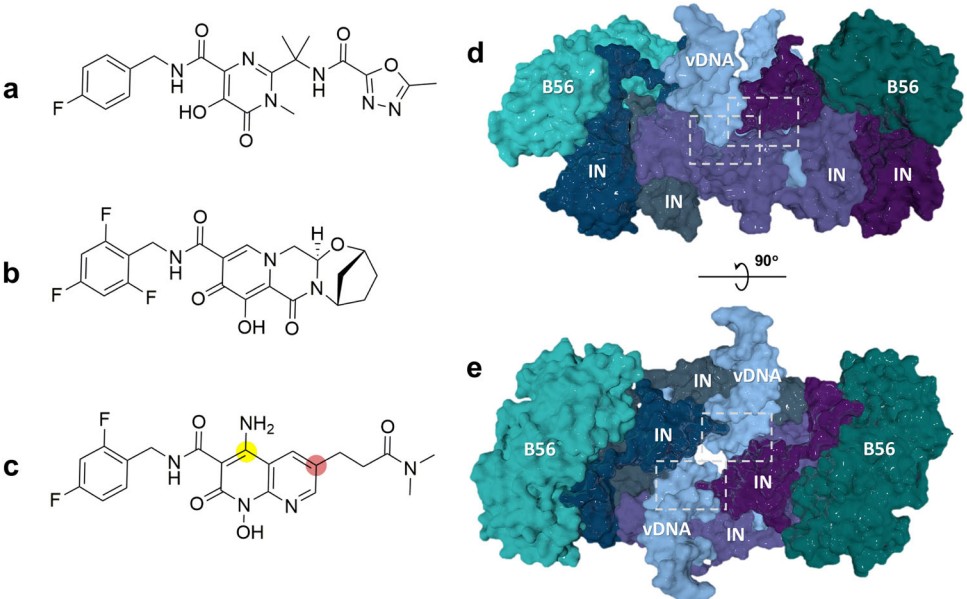

**Fig. 2 Structures of the INSTIs and the STLV-1 Intasome.** Chemical structures of the INSTIs used in the structural studies of binding to the deltaretroviral intasome: **a** Raltegravir. **b** Bictegravir. **c** XZ450. Most frequently substituted positions 4 and 6 are indicated with yellow and pink circles, respectively. **d–e** Overall structure of the STLV-1 intasome: RAL complex solved by cryo-EM. Each colour represents an independent integrase (IN, grey, dark blue, indigo and violet) and B56 (light and dark green) chain. Viral DNA (vDNA) is coloured light blue. Active site regions are marked with white rectangles.

both the particle concentration and the distribution in particle orientation) allowed us to obtain three STLV-1 intasome: INSTI complexes, with FDA-approved INSTIs RAL and BIC, and XZ450, which was the most promising INSTI candidate identified in our screen (Fig. 2a–c and Supplementary Figs. 11–13 and Supplementary Table 6). Cryo-EM maps of the intasome:INSTI complexes were refined to an overall resolution of 3.1 Å in the case of RAL complex, 3.5 Å for BIC and 3.0 Å for XZ450. The local cryo-EM map resolution around the active site region extended to ~2.8 Å, 3.0 Å and 2.5 Å, respectively. The quality of the 3D reconstructions allowed for unambiguous placement and building of the INSTI compounds within the intasome binding pocket (Supplementary Fig. 14). The STLV-1 intasome:INSTI structures showed, as we previously reported[16], a twofold symmetric IN dimer of dimers with the two B56γ molecules that laterally flank the intasome (Fig. 2d, e). All 4 IN chains and their 12 structural domains are resolved remarkably well within the respective cryo-EM maps. We previously reported that all four SLiM motifs in IN are involved in binding the two B56γ molecules. The SLiM peptide belonging to the outer (non-catalytic) IN subunit binds B56γ in the canonical SLiM-binding site. At the same time, the inner (catalytic) IN SLiM engages an additional highly conserved site on B56γ. In the current intasome structures, the mutant E219 residue is seen facing the canonical SLiM-binding site on B56γ in the outer intasome chain and the IN catalytic core domain (CCD) of a neighbouring IN, in the inner IN chain (Supplementary Fig. 15). Putative interactions with neighbouring residues such as K240 or H243 on B56γ likely play a role in the stabilisation provided by the A219E mutation.

**Binding mode of INSTIs to the STLV-1 intasome.** During retroviral integration, the intasome inserts both ends of the viral genome into host DNA in two separate strand transfer reactions. The twofold symmetrical arrangement of retroviral intasomes catalyses the two transesterification events resulting in the concerted integration of both ends of the vDNA into the host DNA. As in most transposases, the catalytic site is defined by the Asp-Asp-Glu catalytic triad (D65, D122 and E158 in HTLV and STLV

INs), which coordinates two catalytically essential $Mg^{2+}$ ions. The metal ions are resolved in the STLV intasome:INSTI structures (Fig. 3a). The binding of RAL to the intasome is largely mediated by strong interactions between the three co-planar oxygen atoms of the drug and the $Mg^{2+}$ cations. Parallel π-π stacking of RAL's fluorobenzyl ring against the penultimate 3'-cytosine in STLV vDNA and interactions between the oxadiazole and Y149 strengthen the binding (Fig. 3b).

As observed in previously reported intasome:INSTI structures[13–15], INSTI binding is accompanied by a flip of the terminal vDNA 3'-adenosine, which vacates a pocket formed between a dC:dG basepair and a $3_{10}$ helix (IN residues 151–155 in STLV and HTLV-1 INs). The pocket becomes occupied by the fluorobenzyl group of the drug, whereas the displaced adenosine base forms parallel π-π stacking interaction with the heterocyclic backbone of the drug (Fig. 3c, d). As shown in Fig. 3d, when compared with the apo form of the STLV-1 intasome, the penultimate vDNA cytosine, which is stacked against the terminal adenine in the apo form, makes a π-π stacking interaction with the halobenzyl ring of RAL; however, the position of the cytidine is very similar in the two structures.

BIC, a second-generation INSTI, lacks the oxadiazole substituent and features an extended scaffold that allows it to span the IN active site (Fig. 4). The interaction with Y149 (Y143 in HIV), which is an important contributor to the binding of RAL, is largely absent. Because second-generation INSTIs do not form strong contacts with Y143 in HIV-1 IN, these compounds can be used to treat HIV-1 infections that involve viruses that carry resistance mutations at this position[23]. This suggests that if an HTLV RAL-resistant Y149 mutant arose, it is very likely that the mutant would be sensitive to BIC. The position of BIC within the binding pocket brings the terminal heterocycle into proximity with the IN/CCD β4-α2 where it has close interactions with the backbone atoms of residues N123 and G124 (Fig. 4 and Supplementary Fig. 16). The corresponding residues of the HIV-1 and SIVrcm INs (N117 and G118) were recently also reported to play a role in the binding of second-generation INSTIs to the primate lentiviral intasomes[14,15].

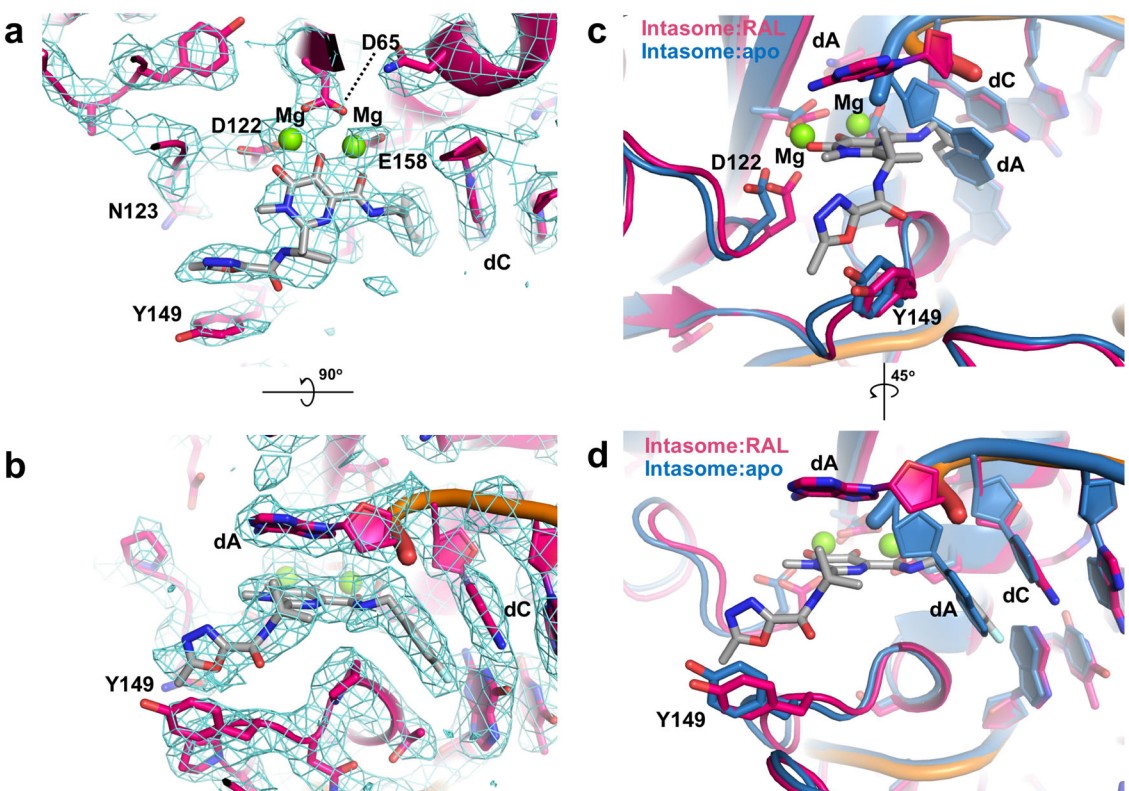

**Fig. 3 Close-up of the active site of STLV-1 intasome (red) in a complex with RAL (grey). a–b** Overall structure of the complex with the cryo-EM map (cyan) overlaid. Magnesium ions are shown as green spheres. **c–d** Superposition of the STLV-1 intasome:RAL complex (red) on the previously reported apo form of the STLV-1 intasome (PDB ID: 6Z2Y; blue). The DNA of the intasome:apo structure is shown in gold. RAL is shown as sticks with the carbon atoms in grey. *dA,* deoxyadenosine; *dC,* deoxycytidine; *RAL,* raltegravir.

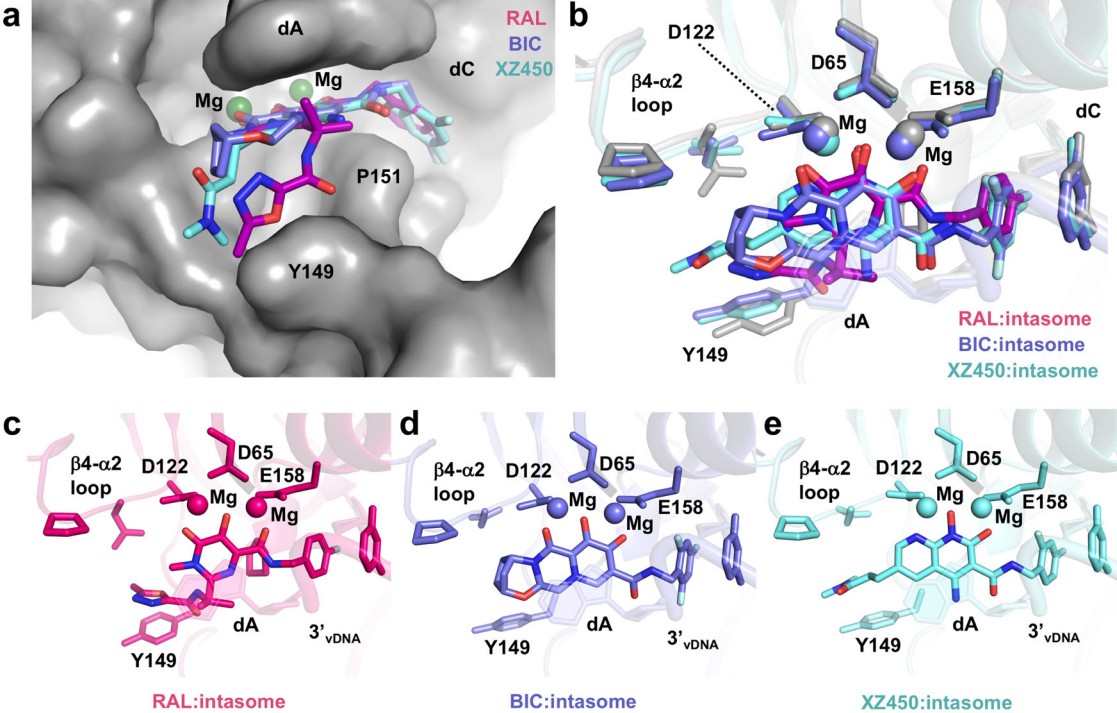

**Fig. 4 STLV-1 intasome:INSTI complex structures. a** View of the active site with the three INSTI structures overlaid binding to the STLV-1 intasome. Magnesium ions are coloured green. For clarity, the true positioning of the INSTIs is shown in the context of the intasome:RAL complex structure only (grey) which is shown in surface view (**a**) and sticks (**b**). **c–e** Active sites of all three complete intasome:INSTI complex structures are shown in cartoon with side chains as sticks. Magnesium ions are coloured according to the colour of the rest of the model. The drugs are shown as sticks, with the carbon atoms of RAL in magenta, BIC in purple and XZ450 in cyan. *dA,* deoxyadenosine; *dC,* deoxycytidine; *RAL,* raltegravir; *BIC,* bictegravir.

**Binding mode of XZ450 to the STLV-1 intasome**. XZ450 represents a class of INSTIs built on a naphthyridine-based backbone. It is one of a chemical library of candidate INSTIs designed to fill the binding pocket to a larger extent and occupy the substrate envelope. It is thought that an inhibitor, which in addition to disabling the correct conformation of the vDNA donor, largely fills the substrate envelope, but stays within the envelope, would be more active against mutant forms of IN[11,14,24]. XZ450 binds to the active site of the STLV intasome in a fashion similar to BIC (Fig. 4). However, due to the presence of the 3-(dimethylamino)-3-oxopropyl substituent at position 6, it reaches much further along the active site. The 6-substituent positions itself equidistantly between the β4-α2 linker and Y149, coming in close contact (<4 Å) with the backbone and side-chain atoms of both regions (Supplementary Fig. 17). Y149 is shifted significantly closer to the drug, compared with its position in the RAL- and BIC-bound intasome structures (Fig. 4b).

**Similarities to other retroviral intasome:INSTI complexes**. Despite the fact that retroviral integrases, in general, share a well-conserved active site, previous structural research has shown that even small changes in the conformation of active site residues can significantly affect the binding and efficacy of INSTIs[14,15]. The mode of binding of some INSTIs has also been shown to differ between the HIV-1 and PFV intasomes[14]. We, therefore, analysed the similarities in STLV-1 and HIV-1 intasomes in the conformations of the residues in the active sites, the positions of Mg$^{2+}$ co-factors and the positions of RAL, BIC and XZ450/XZ419. Since HIV-1 intasome:RAL complex structure is not available, we used the PFV intasome:RAL crystal structure[13] instead. We also used the structure of the complex between HIV-1 intasome and XZ419[14] (a close analogue of XZ450) because the structure of the HIV-1 intasome with XZ450 bound is not available.

The conformations of the bound INSTIs and their surrounding active site residues are very similar within STLV-1, HIV-1 and PFV intasome structures (Fig. 5). Undoubtedly, this is the reason for the high sensitivity of HTLV-1 IN to various INSTIs originally designed to inhibit HIV-1 IN (this and previous[9,25] studies). STLV-1 IN residue Y149 (corresponding to Y143 and Y202 in HIV-1 and PFV, respectively) exhibits the largest difference. The tyrosine side-chain twists, displacing the terminal oxygen atom of Y149 by 4 Å in the STLV-1 intasome structure compared with the HIV-1 structure. However, there are differences in the

conformation of Y149 in different STLV intasome:INSTI complexes (Fig. 4b).

## Discussion

Having evaluated 32 compounds, we identified two small molecules, XZ450 and XZ446, which are two- to fivefold more potent than RAL in blocking HTLV-1 infection in viral culture. Interestingly, EC$_{50}$ values for these compounds in previously reported studies of infection by wild-type HIV-1 did not show an improvement in inhibition over RAL ($4 \pm 2$ nM for RAL compared with $5 \pm 1.3$ nM for XZ450)[11]. The inhibitory potency of XZ446 for HIV-1 ($2 \pm 0.1$ nM)[10] was similar to that for HTLV-1 ($2.57 \pm 0.81$ nM). This illustrates that small changes within the active site can have a significant effect on the binding and efficacy of inhibitors. XZ448 activity was much less potent in its ability to inhibit HIV-1 replication ($263 \pm 52$ nM)[11] similar to our results with HTLV-1. We suspect this could be due to the less efficient uptake of XZ448 by the cells.

The architecture of the deltaretroviral intasome most closely resembles the structure of the PFV intasome, both are IN tetramers[13]. The binding of several classes of INSTIs to the PFV intasome has been structurally characterised and used as a model of INSTI binding to the HIV-1 intasome for the last 10 years[10,11,26–30]. However, it was recently reported by Lyumkis et al.[14] that the binding of some INSTIs can differ between PFV and HIV. For example, the sulfonyl benzyl group of the 6-substituted naphthyridine compound XZ446 (4 f) is rotated by over 90° within the binding pocket of HIV intasome compared with that bound in PFV, resulting in association with a different set of IN residues[14]. This underlines the importance of structural characterisation of INSTI binding specifically for the clinically important virus HTLV-1.

The nearly identical conformation of the INSTIs RAL, BIC and XZ450 bound to STLV-1 IN, HIV-1 IN and PFV IN explains why all three compounds, which were developed to bind to HIV-1 IN, potently inhibit HTLV-1 replication (Fig. 1 and Supplementary Figs. 4 and 5 and Supplementary Table 2)[9,25]. Although the STLV-1 structures show that there is a small change in conformation of residue Y149 depending on which INSTI is bound, this difference is owing to the way the oxadiazole ring of RAL and the 3-(dimethylamino)-3-oxopropyl substituent of XZ450 interact with the tyrosine ring. The shift of the tyrosine side chain is much more pronounced when STLV-1 and PFV intasome:RAL structures are compared (Fig. 5a inset).

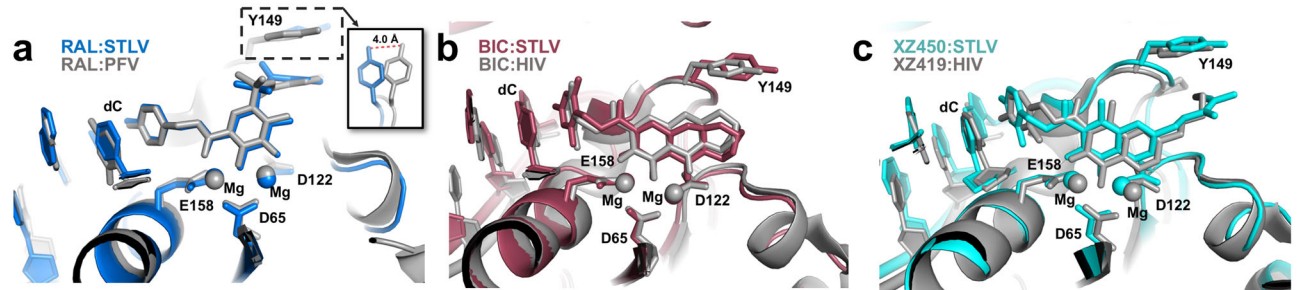

**Fig. 5 Similarities to other retroviral intasome:INSTI complexes.** Comparison between the mode of binding of RAL (**a**), BIC (**b**) and naphthyridine-based INSTIs XZ450 and XZ419 (**c**) to STLV-1 and HIV-1/PFV intasomes. The inset in **a** shows a significant shift in the position of STLV-1 residue Y149 (not visible in the original panel). PFV and STLV-1 IN have 35.8% sequence similarity, RMSD calculations comparing active sites were calculated as described in the Methods. For PFV intasome:RAL versus STLV-1 intasome:RAL the RMSD within the active site is 1.680 Å (214 atoms to 214 atoms). **b** HIV-1 and STLV-1 display 40.43% sequence similarity, RMSD within the active site between the HIV-1 intasome:BIC and STLV-1 intasome:BIC structures is 4.516 Å (238 atoms to 238 atoms). **c** RMSD within the active site for HIV-1 intasome:XZ419 versus STLV-1 intasome:XZ450 is 2.122 Å (228 atoms to 228 atoms). The following PDB models were used: 3OYA (**a**), 6PUW (**b**) and 6V3K (**c**). Superpositions were conducted by the align function in PyMOL and adjusted manually if needed. *dC*, deoxycytidine; *PFV*, prototype foamy virus; *RAL*, raltegravir; *BIC*, bictegravir.

Despite the overall similarity between deltaretroviral and HIV-1 IN active sites, there are several notable differences. Thus, HIV-1 IN residues Q148 and G140 are replaced with S154 and H146 in HTLV-1 and STLV-1 INs. The Q148H/G140S substitutions in HIV-1 IN are well-known resistance mutations to the first-generation INSTIs. Cryo-EM studies of the SIVrcm Q148H/G140S intasome indicated that histidine in position 148 causes displacement of an active site water molecule coordinated by a network of hydrogen bonds involving residues Q148, E152 and D116 in the native intasome[15]. Instead, H148 and S140 hydrogen bond each other. These changes are thought to perturb the binding of INSTIs to the intasome, yet second-generation INSTIs such as BIC are able to escape such resistance by additional interactions further along with the substrate envelope. The complex of BIC with the deltaretroviral intasome shows that there are similar interactions with the backbone of the IN β4–α2 linker (Fig. 4). We were interested in whether the presence of H146 and S154 in HTLV/STLV, albeit in reciprocal positions, would also cause displacement of the active site water molecule and therefore to some extent affect the binding of RAL. As shown in Supplementary Fig. 18, this is clearly not the case as the water molecule is present, coordinated by E158 and D122, and within a reasonable distance (3.2 Å) to S154 to allow hydrogen bonding; the side chain of H146 faces away from the active site.

The structures presented here explain the high efficacy of HIV-1 INSTIs against HTLV-1 and, as such, are of obvious clinical importance. The inhibitory potencies of the many compounds in both the strand transfer and infection assays (Supplementary Table 2) will also provide a foundation for further pharmacological studies of HTLV-1. We have shown here that compound XZ450 is a potent inhibitor of HTLV-1 transmission in vitro. Pharmacokinetic studies are needed to inform us of its potency and clinical applicability. Limited data are available, but it is promising that so far, no resistance mutations have been reported in HTLV-1 patients treated with antiretrovirals for more than one year and experiencing a strong reduction in PVL[31]. Based on our structural data, it is likely that if IN resistance mutations would occur in RAL-treated patients; these mutants would also be less efficiently inhibited by XZ450. For now, BIC appears to be the more favourable HTLV drug candidate, however, there are no data available yet on either the pharmacokinetics of BIC during pregnancy or the safety and efficacy of using BIC in pregnant women with HIV. RAL is the recommended INSTI to suppress viral load in HIV-1 pregnant women and prevent mother-to-child transmission, and no adverse effects have been found[32–34]. Given that individual INSTIs present different drug–drug interaction profiles and have different contraindications[35], the more INSTIs that are available that can efficiently block HTLV-1 transmission, the greater the chance of safely treating patients with INSTIs and reducing HTLV-1 spread. Although the applicability of IN-targeted therapy for the treatment of individuals who are already HTLV-1-infected remains to be verified, there is an urgent need for drugs that can be used for pre-exposure prophylaxis and early post-exposure HTLV treatment. INSTIs are not currently a standard of care for HTLV patients, even in countries where the virus is endemic. We believe our data should encourage clinical studies of INSTIs as drugs that can be used to reduce the spread of HTLV-1. Those particularly in need are pregnant and breastfeeding mothers[6] the partners of those who are already infected with HTLV-1, and those who are in the group of the highest risk of exposure, such as homosexual men and intravenous drug users[36–38].

## Methods

### HTLV-1, STLV-1 IN and PPP2R5C constructs for prokaryotic expression.
For expression of HTLV-1 IN, the previously described pET28a(+)-SUMO-H6P-

STLV-1 MarB43 IN construct[16] was used to express wild-type STLV-1 IN (integrase sequence corresponding to nucleotides 4338-5231 of the STLV-1 MarB43 isolate genome, GenBank ID: AY590142.1). Single-point mutants of STLV-1 MarB43 IN were generated by two-step PCR splicing. The following primer pairs were used for cloning SLTV-1 IN mutants: R49Q (GNM860, GNM770 and GNM861, GNM769), I147V (GNM862, GNM770 and GNM863, GNM769), A219E (GNM864, GNM770 and GNM865, GNM769), P214D (GNM866, GNM770 and GNM867, GNM769). The PCR products were then spliced together using primers GNM770 and GNM769, digested with EcoRI/XhoI and ligated into similarly digested pET28(+)-SUMO-H6P[9]. For the expression of LEDGF/ΔIBD-B56γ, the previously described pET28a(+)SUMO-H6P-LEDGF/ΔIBD-PPP2R5C(11-380) construct was used[9]. For primer sequences, see Supplementary Table 3.

### Recombinant protein expression and purification.
HTLV-1 IN was expressed and purified by affinity, ion exchange and size exclusion chromatography[9,39]. STLV-1 IN, its point mutants, and LEDGF/ΔIBD-B56γ were expressed and purified by affinity, ion exchange and size exclusion chormatography[16].

### Electrophoretic mobility shift assays.
The donor DNA (vDNA) that mimics the 3′-processed LTR of STLV-1 DNA was preassembled by annealing Atto-680-labelled MarB43_U5_S30UP and MarB43_U5_S30B oligonucleotides (Supplementary Table 4) in 100 mM Tris pH 7.4, 400 mM NaCl. In all, 16 µg of STLV-1 IN or its mutants were co-incubated with 38.4 µg of LEDGF/ΔIBD-B56γ at 4 °C for 30 min to allow binding. Samples were then supplemented with 0.5 µL 20 µM Atto680-labelled vDNA, and sufficiently diluted to give a final NaCl concentration of 60 mM. This was immediately placed in a 37 °C heating block for 10 min. The sample was then returned to room temperature (RT) and any precipitate was resolubilised by the addition of NaCl to a final concentration of 1.2 M. After equilibration of the samples at RT for 10 min, the samples were either immediately supplemented with 10 µg/mL heparin or, where stated, were incubated further or additional reagents/INSTIs were added, as described. Samples, supplemented with 10 µg/mL heparin, devoid of loading dye were carefully loaded onto a 3% low melting point agarose gel containing 10 µg/mL heparin and electrophoresed, avoiding overheating. Densitometry analysis was carried out in ImageJ v1.50i. Measurements from at least three independent experiments were taken. Standard deviations and p values were calculated in Prism 9.

### Pull-down assays.
Pull-down assays were conducted as follows[16]: Ni-NTA resin was washed with pull-down buffer containing: 25 mM Tris-HCl pH 7.4, 150 mM NaCl, 20 mM imidazole and 0.5% CHAPS. Beads were then incubated at 4 °C for 2 h with both 10 µg of 6xHis-tagged STLV-1 MarB43 IN (or mutants thereof) and 20 µg of LEDGF/ΔIBD-B56γ, in presence of 10 µg bovine serum albumin and pull-down buffer. The resin was subsequently washed five times with the pull-down buffer. After the last spin at 400 × g for 5 min, the supernatant was gently removed, and the beads boiled in 20 µl 1.5 × SDS loading buffer containing 200 mM imidazole. Samples were centrifuged and 10 µL of the supernatant was analysed on an 11% sodium dodecyl sulphate–polyacrylamide gel electrophoresis gel.

### In vitro integration strand transfer assays.
The donor vDNA that mimics the 3′-processed LTR of HTLV-1 or STLV-1 DNA was preassembled by annealing the U5_S20Q_UP and U5_S20Q_B or MarB43_U5_S30UP or MarB43_U5_S30B oligonucleotides, respectively (Supplementary Table 4) in 100 mM Tris pH 7.4, 400 mM NaCl. For HTLV-1 IN, strand transfer reactions were conducted in a solution of 24.3 mM PIPES pH 6.0, 138.4 mM NaCl, 16.6 mM MgCl₂, 5.8 µM ZnCl₂, 12.7 mM DTT, 0.53 µM donor DNA and 4 µg of HTLV-1 IN[9,39]. In brief, the reaction was incubated at RT for 15 min in the presence of an INSTI or dimethyl sulfoxide (DMSO) control and the reaction was started by the addition of 300 ng of pGEM-9Zf(-) supercoiled plasmid DNA. The reaction was incubated for 1 h at 37 °C and stopped by adding 0.5% SDS and 25 mM ethylenediaminetetraacetic acid (EDTA), pH 8.0.

For STLV-1 IN, the reaction conditions were 25 mM BTP-HCl pH 6, 0.1 µM STLV-1 IN, 2 µM vDNA, 60 mM NaCl, 13.28 mM DTT, 10 mM MgCl₂, 10 µM ZnCl₂. STLV-1 IN and LEDGF/ΔIBD-B56γ were co-incubated at the above conditions at a 2:1 molar ratio, at 4 °C for 30 min. Donor vDNA was then added and the mixture was co-incubated at 37 °C for 10 min. For INSTI dose–response experiments, either the INSTI or DMSO control were then added, the sample was mixed and kept at 4 °C for an additional 15 min. Strand-transfer reactions were initiated by the addition of 300 ng of pGEM-9Zf(-) supercoiled plasmid DNA and stopped after 15 min of 37 °C incubation by adding 0.5% SDS and 25 mM EDTA, pH 8.0.

Proteins were degraded by incubation with 30 µg of proteinase K (Roche) at 37 °C for 1 h. DNA was precipitated at −20 °C overnight in the presence of ethanol supplemented with 20 µg of glycogen and then resuspended in 1.5× agarose loading dye. The DNA products were separated by electrophoresis through 1.5% agarose gel and stained with ethidium bromide and imaged using the BioRad GelDox XR + imager. The data were analysed and the $IC_{50}$ calculated as previously described[9]. In brief, the DNA band that corresponds to the concerted integration product was quantified by densitometry in ImageLab 4.1 (Bio-Rad). All

experiments were done in triplicate. Quantified data points were fitted to dose–response curves in Prism 9. The cumulative standard deviation for each drug was calculated as an average of the upper limit and lower limit values:

$$\text{Lower limit} = \frac{IC_{50}}{10^{\log SE(IC_{50})}}$$

$$\text{Upper limit} = IC_{50} \times 10^{\log SE(IC_{50})}$$

Where SE denotes the standard error computed by Prism 9.

**Cryo-EM grid preparation and data collection**. The STLV-1 intasome was assembled by mixing equimolar (0.03 mmol) quantities of recombinant STLV-1 IN(A219E) and LEDGF/ΔIBD-B56γ and dialysing overnight at 4 °C against 0.5 L of 25 mM Tris-HCl pH 7.4, 200 mM NaCl, 2 mM DTT. This sample was then concentrated at 4 °C, 1935 × g in a 30-kDa MWCO ultrafiltration device (Vivaspin) to a concentration of 0.2 mM. A mixture containing 0.7 mL 20 mM BTP-HCl pH 6, 10 mM CaCl₂, 10 mM DTT, 10 µM ZnCl₂ and 25 µM STLV-1 MarB_U5_S30 double-stranded vDNA (Supplementary Table 4) was placed in a heat block set to 37 °C and incubated for 10 min. The STLV-1 IN(A219E): LEDGF/ΔIBD-B56γ complex was then added, mixed by gently flicking the tube and the tube placed back in the heat block for 10 min. Following incubation, the precipitate was dissolved by the addition of NaCl to a final concentration of 1.2 M, gentle mixing and a further incubation at RT for 15 min. The sample was then purified by size exclusion chromatography[16]. The peak fraction, supplemented with 50 µM INSTI (RAL, BIC, or XZ450) and 20 mM MgSO₄, was incubated at RT for 45 min to allow for the intasome-INSTI complex formation[40]. For all complexes, UltraAuFoil R 1.2/1.3 Au 300-mesh grids (Electron Microscopy Sciences)[41] were glow-discharged for 4 min at 45 mA on an Emitech K100X instrument (Electron Microscopy Sciences) and covered with a layer of GO (Sigma-Aldrich, catalogue #763705)[42] immediately before being used. In brief, to add the layer of GO, a solution of 0.2 mg/ml GO was made in ultrapure water, the suspension was centrifuged at 300 × g for 15 sec to remove aggregates. Following 1 min incubation at RT, the grids were carefully blotted, washed in three drops of water and allowed to air dry for 5 min before use. Four µL of STLV-1 intasome:INSTI complexes were spotted on the graphene side of the coated grid. The grids were incubated for 1 min at 22 °C and 95% humidity, blotted for 2.5 s prior to plunge-freezing in liquid ethane using a Vitrobot Mark IV instrument (Thermo Fisher Scientific).

Cryo-EM data were collected on a 300-keV Titan Krios G3i cryo-electron microscope (Thermo Fisher Scientific) equipped with a Gatan GIF BioQuantum energy filter and a K3 Summit direct electron detector (Gatan). Micrographs were acquired in dose-fractionation mode, at a calibrated magnification corresponding to 1.1 Å per physical pixel (0.55 Å per super-resolution pixel) at the detector. A total electron exposure of 50 e/Å² was fractionated across 41 movie frames. A 20-eV energy slit and a defocus range of −0.7 to −3.6 µm were used for all data collections. The number of micrograph movie stacks acquired for RAL, BIC and XZ450 complexes were 17744, 8502 and 9188, respectively.

**Image processing and 3D reconstruction**. The micrograph movie frames were aligned, binned to the physical pixel size and summed, applying dose weighting as implemented in MotionCor2[43]. Contrast transfer function (CTF) parameters were estimated using Gctf-v1.06[44]. At this stage, images with crystalline ice contamination were discarded; 14,271 (intasome-RAL data set), 6129 (BIC) and 7719 (XZ450) micrographs were retained for further processing. Particles were picked with Gautomatch-v0.53 (http://www.mrc-lmb.cam.ac.uk/kzhang/) using STLV intasome 2D class averages[16], low-pass filtered to 18 Å resolution, as templates. The particles, extracted in Relion-3.1 and binned to a pixel size of 4.4 Å, were subjected to two rounds of reference-free 2D classification in CryoSPARC-2. Particles belonging to well-defined 2D classes (Supplementary Fig. 11) were re-extracted, binned twofold prior to 3D classification in Relion-3.1 into 12 (intasome-RAL complex), 10 (BIC), or 6 (XZ450) classes. Symmetry was not imposed during 3D classification. This procedure yielded a single high-resolution 3D class per data set; particles from the best 3D classes were re-extracted as full-sized images (Supplementary Table 6) and used for 3D reconstruction in Relion-3.1 imposing C2 symmetry. The quality of the maps was further improved by CTF refinement (to estimate beam tilt and per-particles defocus) and Bayesian polishing, as implemented in Relion-3.1. Map resolutions were estimated using the gold-standard Fourier shell correlation 0.143 criterion[45,46] (Supplementary Table 6). Local resolutions of the cryo-EM maps were estimated with Blocres[47] (Supplementary Fig. 12). To aid in the model building process and to prepare figures, the maps were filtered and sharpened using deepEMhancer[48]. For real-space refinements, the cryo-EM maps were sharpened and filtered using density modification procedure in Phenix using default parameters[49].

**Intasome: INSTI model building and refinement**. Density modification was performed under default parameters in Phenix 1.18-3845[50], using half-maps and macromolecular sequence as inputs. The building was initiated with the model of the STLV-1 intasome:B56 complex, reduced to an asymmetric component corresponding to half-intasome. The IN A219E mutation was introduced in the model and the apparent conformational changes of some of the residues and vDNA bases were modelled for in real-space in Coot 0.9.4[51]. RAL and BIC structures were

obtained from existing structures with accession codes 3OYA and 6RWM. The ligands were rigid-body fitted and locally refined. Density for the magnesium atoms was apparent and allowed their unambiguous positioning and water molecules were added to complete the Mg²⁺ coordination sphere. The model was duplicated and rigid-body docked in Chimera 1.12.0[52] to form the missing symmetrical part. The model was manually adjusted, and the final real-space refinement was conducted using Phenix version dev-4142 and the density modified map. Secondary structure restraints and basepair/base stacking definitions based on the model, metal bond, ligand restraints and NCS constraints for the two halves of the symmetric nucleoprotein assembly were used. The quality of the final atomistic model was assessed with MolProbity and EMRinger (Supplementary Table 6).

**Root-mean-square deviation (RMSD) calculation**. All-atom RMSD calculations were performed in Pymol v1.8.0.3 using the align or super function without outlier rejection. For RMSD calculations comparing active sites, all residue atoms and heteroatoms (excluding waters) within a 12 Å distance from the centre of the given INSTI were selected for the calculations.

**Cell-to-cell HTLV-1 infection**. MT-2 and E6.1 Jurkat T-cell lines (ATCC) were used for the experiments. The cells were cultured at 37 °C with 5% CO₂ and maintained in Roswell Park Memorial Institute (RPMI) media supplemented with 10% fetal bovine serum (FBS), 100 U penicillin, 100 µg/mL streptomycin and 0.25 µg/mL fungizone.

The HTLV-1 cell-to-cell infection was performed as previously described[9]; Jurkat T cells were counted and diluted in complete RPMI to a final concentration of 0.25 × 10⁶ cells/ml and pre-treated with INSTI drugs (final concentration ranging from 0.2 pM to 2 µM) or DMSO as a control for 24 h. MT-2 cells were resuspended in RPMI at a concentration of 2 × 10⁶ cells/ml and subjected to a nonlethal γ-radiation dose of 400 Gy. Cells were resuspended at a concentration of 2 × 10⁶ cells/ml in serum-free RPMI supplemented with the corresponding INSTI drug. Co-culture of 0.5 × 10⁶ for each cell line was carried out for 16 h, after which cells were washed in PBS and resuspended in 1 ml of depletion buffer (PBS, 0.1% FBS and 2 mM EGTA). MT-2 cells were depleted from the culture by means of anti-CD25 magnetic beads (DynabeadsTM CD25, Thermo Fisher). Depletion efficiency was confirmed via flow cytometry as described previously[9]. Jurkat cells were cultured for 2 weeks in complete RPMI supplemented with the drugs and genomic DNA was extracted (DNeasy kit, Qiagen) and stored at −20 °C for subsequent analysis. All the experiments were done in triplicate.

**Proviral load determination and Alu-qPCR**. The PVL was measured as previously described[9,53,54]. Jurkat genomic DNA was quantified and diluted to a concentration of 5 ng/µL. Real-time qPCR analysis were conducted on 20 ng of gDNA to measure the level of HTLV-1 tax and gapdh genes (Supplementary Table 5). For the standard curve, the gDNA from an HTLV-1 naturally infected T-cell clone (clone 11.50, courtesy of professor Charles Bangham), which has one copy of tax and two of gapdh, was used. The integrated provirus copy number was calculated as follows: (tax copy numbers)/(2×gapdh copy number)[53], and normalised to the DMSO-treated control which was set to 100%. The PVL data were fitted to a dose–response curve in Prism 7 and the EC₅₀ was calculated[9]. Alu-PCR was performed as previously described in ref. [55]. In brief, the first PCR reaction was done with Phusion polymerase starting from 80 ng of gDNA using the ALU-specific primers (Alu-F/Gag-R). The nested PCR was run diluting the amplified product 1:50. The reaction was done as a qPCR using Gag-specific primers (Gag-R/Gag-F) and the Luna SYBR dye reaction mix (New England Biolabs). The integrated provirus copy numbers were normalised to GAPDH and Jurkat cells treated with DMSO and infected with HTLV-1 were set to 100%. Experiments were done in triplicate.

**INSTI inhibitors**. Raltegravir and BIC were purchased from Adooq Bioscience (USA). The majority of the compounds shown in Supplementary Table 1 including hydrazides: XZ15, XZ45, XZ49; oxoisoindoles: XZ89, XZ90, XZ259; pyridinones: XZ202, XZ220, XZ235, XZ236; diketo acids: XZ319, XZ320 and naphthyridines: XZ351, XZ378, XZ384, XZ413, XZ419, XZ423, XZ426, XZ432, XZ434, XZ439, XZ446, XZ448, XZ450, XZ452, XZ456, XZ460, XZ462 were prepared according to the references indicated in Supplementary Table 1. The preparation of compounds XZ242, XZ256 and XZ420 are discussed in Supplementary Materials and Methods. In brief, pyridinones XZ242 and XZ256 were prepared using our previously reported Pummerer cyclization deprotonation cycloaddition cascade of imidosulfoxides[56,57]. Naphthyridine XZ420 was prepared by a modification of our previous methodology, which included the use of a key Suzuki coupling[10]. Detailed synthesis is presented in Supplementary Materials and Methods.

**Reporting summary**. Further information on research design is available in the Nature Research Reporting Summary linked to this article.

## Data availability
The cryo-EM structures have been deposited with the Protein Data Bank and EMDB and are available under the following identifiers: STLV-1 intasome: XZ450 structure: 7OUF

and EMD-13075; STLV-1 intasome: RAL structure: 7OUG and EMD-13076; and STLV-1 intasome: BIC structure: 7OUH and EMD-13077. Correspondence and requests for materials should be addressed to G.N.M. (g.maertens@imperial.ac.uk). Source data are provided with this paper.

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

## Acknowledgements

We thank Drs A. Purkiss and P. Walker for computer and software support. We thank professor C. Bangham for generously sharing with us clone 11.50. We are grateful to Pavel Afonine and Oleg Sobolev for their support in using Phenix.real_space_refine. This work is supported by the Wellcome Trust (Investigator Award to G.N.M., 107005/Z/15Z). Work in P.C. laboratory is supported by US National Institutes of Health grant P50 AI150481 and the Francis Crick Institute (FC001061), which receives its core funding from Cancer Research UK (FC001061), the UK Medical Research Council (FC001061) and the Wellcome Trust (FC001061). Data were collected at the London Consortium Electron Microscopy Facility at The Francis Crick Institute funded by Wellcome Trust (206175/Z/17/Z to Professor Xiaodong Zhang). X.Z., S.J.S., T.R.B. and S.H.H. are supported by the NIH Intramural Program, Center for Cancer Research, National Cancer Institute and by grants from the NIH AIDS Intramural Targeted Program (IATAP). This article is independent research funded by the National Institute for Health Research (NIHR) Imperial Biomedical Research Centre (BRC). The views expressed in this publication are those of the authors and not necessarily those of the NHS, the National Institute for Health Research or the Department of Health.

## Author contributions

M.S.B. purified all wild-type and mutant proteins, assembled intasomes for biochemical analysis and cryo-EM data collection; M.S.B. and P.C. analysed the cryo-EM data; M.S.B, P.C. and V.E.P. refined the atomistic model; T.V. performed the in vitro integration assays and viral inhibition assays; X.Z.Z. and T.R.B. designed and synthesised the library of chemical compounds; S.J.S. provided guidance for choosing, among the large collection of XZ compounds, which compounds would have a good chance to be effective against HTLV/STLV; A.B.-C. prepared GO carbon and prepared the grids for cryo-EM data collection; N.B.C. collected the cryo-EM data; G.N.M. designed and guided the experiments; M.S.B., G.N.M., P.C. and S.H.H. prepared the manuscript with contributions of all authors.

## Competing interests

The authors declare no competing interests.
