## [Peer Review File · Nature Communications]

REVIEWER COMMENTS

Reviewer #1 (Remarks to the Author):

It has been previously reported that INSTIs can be active against HTLV-1 integration (Rabaaoui et al. 2008, Seegulam and Ratner. 2011, Barski et al. 2019). Nevertheless, in this comprehensive study, Barski et al. investigated 32 different compounds for their activity hindering HTLV-1 integration. The study identified two compounds (XZ446, XZ450) that were 2 to 5 fold more effective than the first-generation inhibitor, RAL. The study also determined the high-resolution structures of the HTLV-1 intasomes bound to three compounds (RAL, BIC, XZ450) and showed their mode of binding. The manuscript is consistent, well organized, and it provides relevant information for the development of more effective inhibitors to treat and reduce the spread of deltaretrovirus infections. However, there are some important considerations to be addressed as follows:

(1) surprisingly, the authors have not included BIC in their experiments in Figure 1. BIC was one of the three data collected, and most of the discussion involves this drug. Although the IC₅₀ and EC₅₀ of BIC in HTLV-1 have been previously determined in Barski MS et al. 2019, it would still be useful to have BIC in Figure 1. The best INSTIs tested in this study are around 2-5 fold more potent than RAL for the HTLV-1 WT. In the Barski MS et. 2019 study, BIC showed 20-30 fold more activity. I do not believe this fact reduces the importance of this work and neither it downplays the relevance and needs in developing better inhibitors. However, it is important to have a clear message that besides the multiple compounds here presented, there are also commercially FDA-approved INSTIs with high inhibitory activity for HTLV-1.

Minor points

(1) Is there a technical reason for the minor differences in buffer constitution for the in vitro strand transfer assays of HTLV-1 and STLV-1? Were they optimized for both systems?

(2) Table S6 is lacking the number of movies collected and particles initially picked per dataset.

(3) Data is missing the map-model FSC plots for the three datasets. They are important to show consistency between the built model and the map used.

(4) In the model of Fig. S11, the E219 residue appears to be ~ 5 Å apart from the residue H60. Based on the sentence "Putative interactions with ...", Is this interaction possible?

(5) It is not clear in the text why the authors mention that graphene oxide (GO) allowed them to collect the cryo-EM data on three STLV-1 intasomes. Was there a problem with particle concentration, orientation, or stability that was benefit by this method?

(6) Moreover, there is no description or reference about how the GO was prepared in methods. The glow-discharge procedure seems a little off at 4 min. This part of the method requires revision and clarification.

(7) Figure 3. There is no description of a gold structure, although there are gold structures shown in the figure.

(8) Figure 5. Is the other Mg hidden in figure Figure S5b?

Reviewer #2 (Remarks to the Author):

An infection with the human T-cell lymphotropic virus type 1 (HTLV-1) can lead to severe health conditions and is thus in the focus of several research initiatives. In their manuscript "Structural basis for binding of integrase strand transfer inhibitors to HTLV-1 intasome inferred from cryo-EM", Barski and co-workers screened a library of novel integrase strand transfer inhibitor (INSTI) candidates and visualized their top inhibitor as well as two known HIV-1 INSTIs by cryo-electron

microscopy. Instead of HTLV-1, STLV-1 is employed as a model for their structural studies. Their results reveal the binding modes of the inhibitors to the active site of STLV-1 integrase and elucidate subtle differences in the coordination environment of the Mg²⁺ ion pair, interacting with the INSTIs. Further, their findings open up new avenues to ultimately treat HTLV-1 infections in the future.

The work is original, well and comprehensively written. Parts of it are of high interest. The experimental work is technically sound and provides strong evidence for the conclusions made. The current state-of the art with respect to literature is implemented. The authors undertake a smart approach to improve accessibility of their sample to cryo-EM by introducing phosphoserine mimetic mutations and thus stabilizing the STLV-1 intasome: INSTI complexes. It is very interesting to see the binding mode of first- and second-generation INSTIs to the deltaretroviral intasome and in addition a novel yet uncharacterized compound, XZ450, with superior binding to STLV-1 and also HTLV-1. This very comprehensive work will seed and alleviate follow up structural studies of inhibitors bound to delta-viral INT complexes. Partially, this work may be transferred to other viral IN complexes and contribute to the enhancement of HTLV-1 integration inhibitors. I have slight concerns regarding the novelty of this study since binding modes of the investigated inhibitors were already structurally characterized in highly homologous other intasomes (as shown in Fig. 5). On the other hand, the subtle differences between these structures and the ones reported in this study are explained in detail in the discussion and the novel STLV-1 / HTLV1 inhibitor XZ450 is a good starting point to develop HTLV-1 inhibitors and thus of considerable medical relevance.

In summary, and despite minor concerns about novelty, I recommend a revision of my points listed below as a prerequisite for publication of this study in Nature Communications.

Major points

The title is a bit misleading since main parts of this work refer to STLV-1. The title should take this into account.

In Supplementary Figure 14, the presence of water molecules is highly speculative, facing the resolutions reported in this study.

Gels of pull down assays are missing for Supplementary Figure 4. In my opinion, bar plots do not suffice here. Likewise, gels of the pull down assays and EMSA are missing in Supplementary Figure 7. I do not consider these gels as Source Data. They should be shown side-by-side to the bar plots.

On page 8, it is hard to conceive why A219 was mutated to E. Even though an introduction of negative charge as phosphoserine mimetic is discussed, the role of this residue needs to be better introduced (like it is the case for P214).

In Figure 3c and d, apo side chain conformations would be beneficial to enable the reader to compare the apo and bound states.

Figure 4 appears too crowded to draw any conclusions or relate binding modes to the text. Maybe a side-by-side comparison is advantageous.

Minor points

On page 3, line 9, quoting "12.7% of HAM/TSP patients reported their quality of life was "worse than death" should be amended. Despite the he missing reference, "worse than death" does not sound very scientific.

On page 4, beginning of the second paragraph, "The target of the INSTIs is the IN active site engaged with viral DNA (vDNA) ends in the context of the nucleoprotein complex termed the intasome" is incomplete. I suggest to add " and" before "ends".

On page 4, when stating “we report high-resolution cryo-EM structures of a deltaretroviral intasome with three INSTIs bound.”, it should be made clearer that it is three separate structures.

On page 5, “Most of the compounds were 4-, 6- and 7-substituted naphthyridines, some of which had previously showed...”, please replace “showed” by “shown”.

On page 8, line 10, replace “pull downs” by “pull down assays”. The same issue for Supplementary figure 4.

On page 8, “Cryo-EM maps of the intasome: INSTI complexes were refined to an overall resolution of 3.1 Å” should be re-phrased since typically in cryo-EM, the atomic model is refined against the map.

On page 9, please add a reference to “The STLV-1 intasome: INSTI structures showed, as we previously reported, a two-fold symmetric IN dimer of dimers with the two B56y molecules that laterally flank the intasome (Fig. 2d-e).”

On page 11, “Similarities to other retroviral intasome: INSTI complexes”, I would advise to also provide r.m.s.d.s and amino acid residue similarities between the different complexes, e.g. HIV, PEV, compared.

I find the last sentence of the discussion a bit irritating, especially without any reference given.

In Figure 5, a manual superpositioning is very unusual.
In the Supplement, synthesis of N-Acetyl-N-(3-chloro-4-fluorobenzyl)-2-(ethylthio)acetamide (3), please specify what “cooled to rt” means, i.e. which temperature exactly. I suggest to specify a value for rt in the beginning.

Please provide the r.m.s.d. for Supplementary Figure 5, either in the figure caption or main text.

I would recommend to add dashed lines for Mg²⁺ coordination in Supplementary Figures 11,12.

Reviewer #3 (Remarks to the Author):

The manuscript “Structural basis for binding of integrase strand transfer inhibitors to HTLV-1 intasome inferred from cryo-EM” by Barski et al. reports the naphthyridine moiety containing compound XZ450 as a potent inhibitor of HTLV-1 virus and intasome strand-transfer. CryoEM structures of two delta retroviruses HTLV-1 and STLV-1 have been reported recently including one by the research laboratories of Cherepanov and Maertens. In the current study, they have reported the cryoEM structures of three INSTIs (RAL, BIC, and XZ450) in complex with A219E mutant STLV-1 integrase. A219E mutation improved the stability of the INSTI complexes for the cryoEM study, and the mutation has no noticeable impact on the activity or inhibition. As expected, the binding of inhibitors to STLV IN closely resembles the binding of INSTIs to HIV and PFV INs, and the binding is highly conserved. XZ450 and BIC bind STLV-1 integrase in a similar fashion, however, the substituent at position 6 of XZ450 interacts with Y149, analogous to the interaction of RAL. In summary, the current study establishes a common conserved mode of binding of INSTIs to retroviral integrases and identifies subtle differences that may be important in INSTI design. The manuscript is well written.

Major Comments:

1. As per the authors’ earlier publication Barski et al. 2019, Bictegravir has a favorable antiviral profile against HTLV-1. Then, why the authors in this manuscript advocate for XZ450 as a potential drug candidate?
2. Based on the binding modes of INSTIs to HIV and STLV INs, can the authors comment on the emergence of potential resistance mutation in HTLV IN and their impact on INSTIs? For example, mutation of Y149 that causes RAL resistance in HIV is likely to cause resistance to XZ450. One may argue that based on the potential of the emergence of Y149 mutation, BIC is a favorable

HTLV drug candidate than XZ450.

3. An updated table comparing the potency of key INSTIs to HTLV vs HIV INs, like in Barski et al. 2019, will be informative.

Minor comments:

1. Real-space correlation coefficient and B-factors for individual inhibitors in Table S6 will be informative.
2. Page 14: Figure 5a inset?

Response to Reviewer's comments.

We are grateful for the comments raised by each of the Reviewers. This has given us the chance to improve the manuscript and include further clarification where this was needed.

Reviewer #1 (Remarks to the Author):

It has been previously reported that INSTIs can be active against HTLV-1 integration (Rabaaoui et al. 2008, Seegulam and Ratner. 2011, Barski et al. 2019). Nevertheless, in this comprehensive study, Barski et al. investigated 32 different compounds for their activity hindering HTLV-1 integration. The study identified two compounds (XZ446, XZ450) that were 2 to 5 fold more effective than the first-generation inhibitor, RAL. The study also determined the high-resolution structures of the STLV-1 intasomes bound to three compounds (RAL, BIC, XZ450) and showed their mode of binding. The manuscript is consistent, well organized, and it provides relevant information for the development of more effective inhibitors to treat and reduce the spread of deltaretrovirus infections. However, there are some important considerations to be addressed as follows:

(1) surprisingly, the authors have not included BIC in their experiments in Figure 1. BIC was one of the three data collected, and most of the discussion involves this drug. Although the IC₅₀ and EC₅₀ of BIC in HTLV-1 have been previously determined in Barski MS et al. 2019, it would still be useful to have BIC in Figure 1. The best INSTIs tested in this study are around 2-5 fold more potent than RAL for the HTLV-1 WT. In the Barski MS et. 2019 study, BIC showed 20-30 fold more activity. I do not believe this fact reduces the importance of this work and neither it downplays the relevance and needs in developing better inhibitors. However, it is important to have a clear message that besides the multiple compounds here presented, there are also commercially FDA-approved INSTIs with high inhibitory activity for HTLV-1.

We appreciated this comment. It is important to emphasize the fact that there are commercially FDA-approved INSTIs with high inhibitory activity for HTLV-1. We have included the IC₅₀ and EC₅₀ values for BIC in Figure 1b and 1c and included the EC₅₀ value in Table S2.

Minor points

(1) Is there a technical reason for the minor differences in buffer constitution for the in vitro strand transfer assays of HTLV-1 and STLV-1? Were they optimized for both systems?

Yes. The strand transfer assays were optimised for both HTLV-1 and STLV-1 integrase, hence the slight difference in buffer constitution (BTP vs PIPES).

(2) Table S6 is lacking the number of movies collected and particles initially picked per dataset.

We have now included this in the Supplementary Table S6.

(3) Data is missing the map-model FSC plots for the three datasets. They are important to show consistency between the built model and the map used.

We have now included it as Supplementary Figure S10. Numbering of the remaining supplementary figures have thus been adjusted and are highlighted throughout the text.

(4) In the model of Fig. S11, the E219 residue appears to be ~ 5 Å apart from the residue H60. Based on the sentence "Putative interactions with ...", Is this interaction possible?

IN E219 and H60 are actually over 6 Å apart. We included the second panel to show this possible interaction interface, but we agree it is more likely that the interaction between IN A219E and B56 K240/H243 is the reason for the increased stability of the complex. We have changed the sentence accordingly to: "Putative interactions with neighbouring residues such as K240 or H243 on B56 γ likely play a role in the stabilisation provided by the A219E mutation."

(5) It is not clear in the text why the authors mention that graphene oxide (GO) allowed them to collect the cryo-EM data on three STLV-1 intasomes. Was there a problem with particle concentration, orientation, or stability that was benefit by this method?

The benefit of using GO grids was twofold: First it allowed the concentration of the particles on the grid, second the distribution of the orientation of the particles was much greater. We have now amended the text to include this information.

(6) Moreover, there is no description or reference about how the GO was prepared in methods. The glow-discharge procedure seems a little off at 4 min. This part of the method requires revision and clarification.

We carefully optimised glow discharging of the UltraAuFoil grids and found that 4 min at 45 mA using the Emitech K1000X instrument worked best. This is similar to what was previously reported (5 min at 40 mA but using a different instrument) in the manuscript we refer to in the text. We have further expanded the description of how the GO was applied to the grids: "Briefly, to add the layer of graphene oxide (GO), a solution of 0.2 mg/ml GO was made in water, the suspension was centrifuged at 300 g for 15 sec to remove aggregates. Following 1 min incubation at room temperature, the grids were carefully blotted, washed in drops of water and allowed to air dry for 5 min before use."

(7) Figure 3. There is no description of a gold structure, although there are gold structures shown in the figure.

We have now changed the caption as follows: "... The DNA of the intasome:apo structure is shown in gold. "

(8) Figure S5. Is the other Mg hidden in figure Figure S5b?

In the apo structures only one Mg²⁺ was bound by the intasome. We have now clarified this in the figure caption as well.

Reviewer #2 (Remarks to the Author):

An infection with the human T-cell lymphotropic virus type 1 (HTLV-1) can lead to severe health conditions and is thus in the focus of several research initiatives. In their manuscript "Structural basis for binding of integrase strand transfer inhibitors to HTLV-1 intasome inferred from cryo-EM", Barski and co-workers screened a library of novel integrase strand transfer inhibitor (INSTI) candidates and visualized their top inhibitor as well as two known HIV-1 INSTIs by cryo-electron microscopy. Instead of HTLV-1, STLV-1 is employed as a model for

their structural studies. Their results reveal the binding modes of the inhibitors to the active site of STLV-1 integrase and elucidate subtle differences in the coordination environment of the Mg²⁺ ion pair, interacting with the INSTIs. Further, their findings open up new avenues to ultimately treat HTLV-1 infections in the future.

The work is original, well and comprehensively written. Parts of it are of high interest. The experimental work is technically sound and provides strong evidence for the conclusions made. The current state-of the art with respect to literature is implemented. The authors undertake a smart approach to improve accessibility of their sample to cryo-EM by introducing phosphoserine mimetic mutations and thus stabilizing the STLV-1 intasome:INSTI complexes. It is very interesting to see the binding mode of first- and second-generation INSTIs to the deltaretroviral intasome and in addition a novel yet uncharacterized compound, XZ450, with superior binding to STLV-1 and also HTLV-1. This very comprehensive work will seed and alleviate follow up structural studies of inhibitors bound to delta-viral INT complexes. Partially, this work may be transferred to other viral IN complexes and contribute to the enhancement of HTLV-1 integration inhibitors. I have slight concerns regarding the novelty of this study since binding modes of the investigated inhibitors were already structurally characterized in highly homologous other intasomes (as shown in Fig. 5). On the other hand, the subtle differences between these structures and the ones reported in this study are explained in detail in the discussion and the novel STLV-1 / HTLV1 inhibitor XZ450 is a good starting point to develop HTLV-1 inhibitors and thus of considerable medical relevance.

In summary, and despite minor concerns about novelty, I recommend a revision of my points listed below as a prerequisite for publication of this study in Nature Communications.

Major points

The title is a bit misleading since main parts of this work refer to STLV-1. The title should take this into account.

We have now changed the title to: Structural basis for the inhibition of HTLV-1 integration inferred from cryo-EM deltaretroviral intasome structures.

In Supplementary Figure 14, the presence of water molecules is highly speculative, facing the resolutions reported in this study.

The presence of water molecules in the active sites of retroviral intasomes is regarded as crucial for their catalytic activity. Cook et al. have conducted a thorough study on the influence of active site water positioning and INSTI binding. In our maps we see weak but resolvable density which overlaps very well with the positioning of the waters resolved in the structures of SIVrcm (Cook et al. Science, 2020). We understand the limitations of the overall resolution of our structures yet appreciate the highly-ordered structure of the intasome active site gives rise to a much higher local resolution in that region and is what we believe makes it possible to resolve the weak but clearly present water density at this site. We have amended this figure (now called Supplementary Figure S15) to show this more clearly.

Gels of pull down assays are missing for Supplementary Figure 4. In my opinion, bar plots do not suffice here. Likewise, gels of the pull down assays and EMSA are missing in Supplementary Figure 7. I do not consider these gels as Source Data. They should be shown side-by-side to the bar plots.

We have now included images of pull-down assays and EMSAs in Supplementary Figure S7.

On page 8, it is hard to conceive why A219 was mutated to E. Even though an introduction of negative charge as phosphoserine mimetic is discussed, the role of this residue needs to be better introduced (like it is the case for P214).

We have explained the underlying rationale in the modified text as follows:

“Moreover, a previous report showed that peptides in which additional acidic residues (Glu) were added immediately following the SLiM sequence, had significantly higher affinity for B56²². We thus mutated IN Ala219 into Glu. Indeed, approximately two-fold more B56 γ was recovered...”

In Figure 3c and d, apo side chain conformations would be beneficial to enable the reader to compare the apo and bound states.

We have now added apo side chains to Figure 3 panels c and d.

Figure 4 appears too crowded to draw any conclusions or relate binding modes to the text. Maybe a side-by-side comparison is advantageous.

We have now amended the figure: showing individual intasome:INSTI structures as well as a panel with the overlay.

Minor points

On page 3, line 9, quoting “12.7% of HAM/TSP patients reported their quality of life was “worse than death” should be amended. Despite the missing reference, “worse than death” does not sound very scientific.

This observation comes from the same study that reported that otherwise asymptomatic carriers do suffer from a range of issues that severely impact their quality of life. “Worse than death” is based on the Euroqol five dimension questionnaire (EQ-5D), where scores vary from 0 (death) to 1 (best state of health). Values below zero can be obtained and represent quality of life which is worse than death (EuroQol Research Foundation. EQ-5D-3L user Guide [Internet]. 2018. Available: www.euroqol.org

We have now changed the sentence as follows:

“12.7% of HAM/TSP patients reported their quality of life was worse than death, based on the Euroqol five dimension questionnaire (EQ-5D)⁵.”

On page 4, beginning of the second paragraph, “The target of the INSTIs is the IN active site engaged with viral DNA (vDNA) ends in the context of the nucleoprotein complex termed the intasome” is incomplete. I suggest to add “ and” before “ends”.

We are referring to the vDNA ends that are engaged by the IN active site. The sentence is therefore correct.

On page 4, when stating “we report high-resolution cryo-EM structures of a deltaretroviral intasome with three INSTIs bound.”, it should be made clearer that it is three separate structures.

We have now changed the sentence to: "...we report three high-resolution cryo-EM structures of a deltaretroviral intasome bound to three different INSTIs."

On page 5, "Most of the compounds were 4-, 6- and 7-substituted naphthyridines, some of which had previously showed...", please replace "showed" by "shown".

Corrected.

On page 8, line 10, replace "pull downs" by "pull down assays". The same issue for Supplementary figure 4.

Corrected.

On page 8, "Cryo-EM maps of the intasome:INSTI complexes were refined to an overall resolution of 3.1 Å" should be re-phrased since typically in cryo-EM, the atomic model is refined against the map.

In cryo-EM the map (structure) is refined and then the atomic model is fitted and refined against the map. The sentence is therefore correct.

On page 9, please add a reference to "The STLV-1 intasome:INSTI structures showed, as we previously reported, a two-fold symmetric IN dimer of dimers with the two B56γ molecules that laterally flank the intasome (Fig. 2d-e)."

Included.

On page 11, "Similarities to other retroviral intasome:INSTI complexes", I would advise to also provide r.m.s.d.s and amino acid residue similarities between the different complexes, e.g. HIV, PEV, compared.

We have now included this information in the Figure legend and have included a paragraph in the Methods section which explains how the RMSD calculations were done. They read now as follows:

Root-mean-square-deviation (RMSD) calculation

All-atom RMSD calculations were performed in Pymol using the align or super function without outlier rejection. For RMSD calculations comparing active sites, all residue atoms and heteroatoms (excluding waters) within a 12Å distance from the centre of the given INSTI were selected for the calculations.

"Fig. 5 | Comparison between the mode of binding of RAL (a), BIC (b) and naphthyridine-based INSTIs XZ450 and XZ419 (c) to STLV-1 and HIV-1/PFV intasomes. The inset in (a) shows a significant shift in the position of STLV-1 residue Y149 (not visible in the original panel). PFV and STLV-1 IN have 35.8% sequence similarity, RMSD calculations comparing active sites were calculated as described in the Methods. For PFV intasome:RAL versus STLV-1 intasome:RAL the RMSD within the active site is 1.680 Å (214 atoms to 214 atoms). **b**, HIV-1 and STLV-1 display 40.43% sequence similarity, RMSD within the active site between the HIV-1 intasome:BIC and STLV-1 intasome:BIC structures is 4.516 Å (238 atoms to 238 atoms). **c**, RMSD within the active site for HIV-1 intasome:XZ419 versus STLV-1 intasome:XZ450 is 2.122 Å (228 atoms to 228 atoms). The following PDB models were used: 3OYA (a), 6PUW (b) and 6V3K (c). Superpositions were conducted by the align function in PyMOL and adjusted manually if needed. dC, deoxycytidine."

I find the last sentence of the discussion a bit irritating, especially without any reference given.

References have now been included.

In Figure 5, a manual superpositioning is very unusual.

Corrected now the caption in the revised manuscript.

In the Supplement, synthesis of N-Acetyl-N-(3-chloro-4-fluorobenzyl)-2-(ethylthio)acetamide (3), please specify what “cooled to rt” means, i.e. which temperature exactly. I suggest to specify a value for rt in the beginning.

We have now included this in the General Synthetic Procedures paragraph as follows: “Wherever mentioned, room temperature (rt) was around 22°C.”

Please provide the r.m.s.d. for Supplementary Figure 5, either in the figure caption or main text.

We have now included this information in the figure legend.

I would recommend to add dashed lines for Mg²⁺ coordination in Supplementary Figures 11,12.

In Supplementary Figure S11 there no Mg²⁺ is present. We have however included dashed lines in Supplementary Figure S12. Please note that, since an additional figure was added (Figure S10), the numbering has now changed to resp. Supplementary Figures S12 and S13.

Reviewer #3 (Remarks to the Author):

The manuscript “Structural basis for binding of integrase strand transfer inhibitors to HTLV-1 intasome inferred from cryo-EM” by Barski et al. reports the naphthyridine moiety containing compound XZ450 as a potent inhibitor of HTLV-1 virus and intasome strand-transfer. CryoEM structures of two delta retroviruses HTLV-1 and STLV-1 have been reported recently including one by the research laboratories of Cherepanov and Maertens. In the current study, they have reported the cryoEM structures of three INSTIs (RAL, BIC, and XZ450) in complex with A219E mutant STLV-1 integrase. A219E mutation improved the stability of the INSTI complexes for the cryoEM study, and the mutation has no noticeable impact on the activity or inhibition.

As expected, the binding of inhibitors to STLV IN closely resembles the binding of INSTIs to HIV and PFV INs, and the binding is highly conserved. XZ450 and BIC bind STLV-1 integrase in a similar fashion, however, the substituent at position 6 of XZ450 interacts with Y149, analogous to the interaction of RAL. In summary, the current study establishes a common conserved mode of binding of INSTIs to retroviral integrases and identifies subtle differences that may be important in INSTI design. The manuscript is well written.

Major Comments:

1. As per the authors’ earlier publication Barski et al. 2019, Bictegravir has a favorable antiviral profile against HTLV-1. Then, why the authors in this manuscript advocate for XZ450 as a potential drug candidate?

See answer below.

2. Based on the binding modes of INSTIs to HIV and HTLV INs, can the authors comment on the emergence of potential resistance mutation in HTLV IN and their impact on INSTIs? For example, mutation of Y149 that causes RAL resistance in HIV is likely to cause resistance to XZ450. One may argue that based on the potential of the emergence of Y149 mutation, BIC is a favorable HTLV drug candidate than XZ450.

We appreciate this constructive criticism. We have now included an extra paragraph in the Discussion which reads as follows:

“We have shown here that the new compound XZ450 is a potent inhibitor of HTLV-1 transmission *in vitro*. Pharmacokinetic studies are needed to inform us of its potency and clinical applicability. Limited data are available, but it is promising that so far, no resistance mutations have been reported in HTLV-1 patients treated with antiretrovirals for more than one year and experiencing a strong reduction in PVL³². Based on our structural data, it is likely that if IN resistance mutations would occur in RAL treated patients; these mutants would also be less efficiently inhibited by XZ450. For now, BIC appears to be the more favourable HTLV drug candidate, however, there are no data available yet on either the pharmacokinetics of BIC during pregnancy or the safety and efficacy of using BIC in pregnant women with HIV. RAL is the recommended INSTI to suppress viral load in HIV-1 pregnant women and prevent mother-to-child transmission, and no adverse effects have been found³³⁻³⁵. Given that individual INSTIs present different drug-drug interaction profiles and have different contraindications³⁶, the more INSTIs that are available that can efficiently block HTLV-1 transmission, the greater the chance of safely treating patients with INSTIs and reducing HTLV-1 spread.”

3. An updated table comparing the potency of key INSTIs to HTLV vs HIV INs, like in Barski et al. 2019, will be informative.

We have now included the information in Supplementary Table S2.

Minor comments:

1. Real-space correlation coefficient and B-factors for individual inhibitors in Table S6 will be informative.

The real-space correlation coefficients were present in Table S6, we have now also included the B-factors for the individual inhibitors.

2. Page 14: Figure 5a inset?

We added a black circumference to the inset to enhance its visibility.

REVIEWERS' COMMENTS

Reviewer #1 (Remarks to the Author):

The Authors have properly addressed my former questions, and I am supportive of the publication of this manuscript after clarification of a point in Fig. 1b-c. Why do the EC50 values for BIC and XZ448 are so off compared with their relative values obtained from IC50?

Reviewer #2 (Remarks to the Author):

One last issue is that it would be advantageous to color the water molecules in Suppl. Fig. 15 different than the oxygens (e.g. that of glutamate). All of my previous points raised have been addressed convincingly and I recommend publication in Nature Communications.

Reviewer #3 (Remarks to the Author):

The authors have adequately addressed the reviewers' comments.

REVIEWERS' COMMENTS

We are grateful the referees are supportive of publication of our manuscript provided we addressed the questions below. Please, find our answers highlighted in blue.

Reviewer #1 (Remarks to the Author):

The Authors have properly addressed my former questions, and I am supportive of the publication of this manuscript after clarification of a point in Fig. 1b-c. Why do the EC50 values for BIC and XZ448 are so off compared with their relative values obtained from IC50?

For XZ448 the EC50 is much higher compared to the IC50 value – this was also previously observed for HIV-1. As, mentioned in the first paragraph of the discussion, we suspect this is due to the inefficient uptake of XZ448 by the cells compared to the other drugs. For BIC, the opposite is true, the EC50 is much lower than expected compared to the IC50 value and here we suspect (as we reported in our *Frontiers in Microbiology* paper, Barski M. et al 2019 where we first described the use BIC to block HTLV-1 infection), that BIC is much more efficiently taken up compared to raltegravir e.g.

We have now included this in the Results section of the manuscript as follows: “As shown previously⁹, although bictegrovir (BIC) is only about 2-fold more efficient in blocking HTLV-1 integrase strand transfer *in vitro*, BIC potently blocks HTLV-1 transmission in tissue culture, exhibiting a 20-fold improvement in EC₅₀ (0.3 ± 0.173 nM⁹) compared to RAL. We suspect this is due to a more efficient uptake of BIC than RAL by Jurkat T cells⁹.”

Reviewer #2 (Remarks to the Author):

One last issue is that it would be advantageous to color the water molecules in Suppl. Fig. 15 different than the oxygens (e.g. that of glutamate). All of my previous points raised have been addressed convincingly and I recommend publication in *Nature Communications*.

We have changed the colour of the water molecule to turquoise to distinguish it from the oxygen atoms.

Reviewer #3 (Remarks to the Author):

The authors have adequately addressed the reviewers' comments.